# The Impact of COVID-19 on the Health-Related Behaviours, Mental Well-Being, and Academic Engagement of a Cohort of Undergraduate Students in an Irish University Setting

**DOI:** 10.3390/ijerph192316096

**Published:** 2022-12-01

**Authors:** Elaine Sheedy O’Sullivan, Karrie-Marie McCarthy, Cian O’Neill, Janette Walton, Lisa Bolger, Andrea Bickerdike

**Affiliations:** 1Department of Sport, Leisure & Childhood Studies, Muster Technological University, Bishopstown, T12 P928 Cork, Ireland; 2Department of Biological Sciences, Muster Technological University, Bishopstown, T12 P928 Cork, Ireland

**Keywords:** COVID-19, university students, health-related behaviours, mental well-being, academic engagement

## Abstract

Given the well-established impact of COVID-19 on university students’ health and lifestyle parameters, the current study sought to investigate these impacts within an Irish university setting. A cross-sectional design was employed, with a 68-item questionnaire instrument disseminated to all Year 2 undergraduate students in the host institution (N = 2752), yielding a 9.7% response rate (*n* = 266). This questionnaire elicited students’ self-reported changes to health-related behaviours, mental well-being and academic engagement across 4 defined time-points: (T0: prior to COVID-19, T1: initial onset of COVID-19, T2: during COVID-19, and T3: time of data collection). Many items were adapted from previous Irish research and additional validated scales included the Alcohol Use Disorders Identification Test (AUDIT-C) and the World Health Organisation’s Well-being scale (WHO-5). Key findings revealed that at T1, substantially more males reported ‘*good*/*very good*’ general health than females (76.3% vs. 70.8%), while physical activity patterns followed a similar trend at both T0 (80% vs. 66.1%) and T1 (66.7% vs. 61%). A total of 78.4% of participants reported a body mass gain from T0 to T3, thus reflecting the reduced physical activity levels and compromised nutritional patterns across this period. Worryingly, AUDIT-C scale data revealed hazardous drinking habits were evident in both males and females, while fruit and vegetable intake, physical activity levels, and mental well-being among this cohort remained notably sub-optimal. Ratings of positive academic engagement also decreased substantially between T0 (90.3%) and T3 (30.4%). These findings substantiate the rationale for tailored health promotion interventions in university settings to support students’ transition back to traditional programme delivery and, of equal importance, to improve general health and well-being post-COVID-19 within this cohort.

## 1. Introduction

‘Coronavirus’ (COVID-19) is a highly infectious viral pathogen that emerged worldwide in 2019 and is caused by the SARS-CoV-2 virus [1]. To date, the global incidence of COVID-19 has necessitated substantial mitigation measures across all seven continents, including ‘stay-at-home’ orders (also referred to as localised lockdown period/measures), travel restrictions, severe constraints on social gatherings, and an unprecedented disruption to the traditional delivery of fundamental societal activities such as education, elective medical procedures, entertainment, and sporting activities [1]. Social distancing measures (e.g., staying at home, avoiding crowded areas, using no-contact greetings) and physical distancing (e.g., minimum distance from others outside your household two meters, limits on number of people allowed to gather) led to the closure of many non-essential services in addition to schools and universities. Moreover, there is evidence to suggest that such severe, yet essential, measures resulted in a deluge of detrimental effects on the health and well-being of students within these educational settings [2].

Notwithstanding the direct mortality and morbidity rate, with over 6.3 million global deaths reported to date [3], COVID-19 also indirectly impacted a multitude of population health and well-being parameters [4]. The stay-at-home measures, which were put in place to combat the spread of the virus, have particularly impacted upon health-related behaviours such as reduced physical activity (PA) and increased sedentary time (ST). The aforementioned stay-at-home measures also posed a universal threat to mental well-being [5]. There have been several studies advocating for an exploration of the impact that COVID-19 had on mental well-being, but also public health; particularly in terms of lifestyle behaviours, which may inform efforts to address such adverse trends and the associated modifiable behavioral risk factors [6,7].

A large-scale international study within thirty-five academic organisations (comprised of both research institutes and universities) across Europe, North Africa, Western Asia and America (*N* = 10,477, 55.1% aged 18–35) was conducted to examine the impact of COVID-19 on health and well-being metrics such as PA, diet, social participation, sleep, technology use, and participants’ need for psychosocial support [8]. The authors reported that COVID-19 negatively affected mental well-being and emotional state, while a greater proportion of those who reported a deterioration in their mental state (9.4%) also reported poor dietary behaviours and physical inactivity from ‘before’ the stay-at-home period to ‘during’ the stay-at-home period. A similar large-scale study conducted amongst a sample of 10,082 young adults in China (aged 15–28), reported significant changes in dietary patterns (*p* < 0.05), including a notable decrease in the consumption of fresh fruit (12.4%) and vegetables (7.2%) [9]. Further, in a questionnaire-based study conducted during COVID-19 to examine the health and well-being of adolescents and young adults in the United States (US), more than 60% reported that COVID-19 had affected their eating behaviours and mood state either ‘a little’ or ‘a lot’ [10].

In the United Kingdom (UK), a longitudinal study conducted pre-COVID-19 in October 2019 and repeated during the initial stay-at-home period (April to May 2020), examined the effects of the COVID-19 pandemic on the mental well-being amongst 302, 1st and 2nd Year university students. A comprehensive self-reported questionnaire incorporating validated scales such as Alcohol Use Disorders Identification Test-Concise (AUDIT-C), anxiety and depression (Hospital Anxiety and Depression Scale [HADS]), sleep (Pittsburgh Sleep Quality Index [PSQI]), and mental well-being (Warwick Edinburgh Mental Well-being scale [WEMWBS]) was utilized [4]. Key findings revealed a significant increase in depressive scores (HADS), in addition to concurrent decreases in well-being scores (WEMWBS) across the 2 timepoints. Interestingly, participants reported a significant decrease in alcohol use (AUDIT-C) during the stay-at-home period in comparison to before the onset of the COVID-19 pandemic. In addition, they exhibited no significant changes to their sleep quality (PSQI) or anxiety between the pre-COVID-19 baseline test period and the subsequent follow up test period (i.e., the stay-at-home period) [4].

Further concerning trends have emerged in recent international literature, indicative of the cross-cutting impact of the COVID-19 pandemic on habitual health and lifestyle parameters of university student cohorts. In this regard, studies have demonstrated a reduction in access to healthy food options, irregular dietary habits and significant increases in alcohol consumption [9,11]. Sidebottom et al. [12] conducted a questionnaire-based study among 291 US university students, examining self-report measures of PA levels, sedentary time (ST) and dietary habits (i) before campus closures in March 2020, and (ii) during the stay-at-home periods that followed. In total, 80% (*n* = 233) of participants reported changes in their dietary habits during the stay-at-home period, including increased meal consumption in the home, reduction in fruit consumption, no change in vegetable consumption, and an increase in alcohol consumption. 

In line with global trends, the incidence of COVID-19 in Ireland also resulted in the implementation of stringent public health measures in March 2020 [12,13]. The 2021 ‘Healthy Ireland Survey’, which constitutes the most recent wave of representative data gathered under the remit of the Irish Government’s National Health Promotion Strategy, has provided an invaluable insight into the overarching impact of COVID-19 on health parameters and lifestyle behaviours of the general Irish population [14]. Data were gathered using telephone interviews from 7454 randomly selected individuals, aged 15 years and over, between March 2020 and October 2021. These interviews gathered data pertaining to general health, mental well-being, diet, nutrition, body habitus perceptions, drinking habits, and social connectedness. Worryingly, 81% of respondents (*n* = 6038) reported reduced ‘social connectedness’ as a consequence of COVID-19, thus consolidating the findings of Sidebottom et al. [12] and Schepis et al. [13]. Age-stratified analyses indicated that in terms of drinking habits, 44% of males and 16% of females between the age of 15–24 reported a greater prevalence of binge drinking on a typical drinking occasion during COVID-19. Furthermore, in line with the findings of reduced mental well-being by Evans et al. [4], 45% of participants aged between 15–24 reported a decline in their mental well-being since the beginning of the COVID-19 pandemic, with 13% stating that it had declined to a large degree [14].

In an Irish context, although the impact of the COVID-19 pandemic on the health and lifestyle metrics of the general population has been explored in the above mentioned ‘Healthy Ireland’ study [14] (Department of Health, 2021), there remains a dearth of evidence pertaining specifically to university student cohorts. Although Bickerdike et al. [15] examined the health and well-being parameters of 2267 students in an Irish university prior to the onset of COVID-19, there is a paucity of Irish research exploring the impact of the pandemic on university students, of which there are 245,000 across the island of Ireland [16].

From an academic engagement perspective, the onset of social distancing measures in addition to remote educational delivery models within global universities may have directly contributed to an increase in students’ perceptions of being isolated from peers and teaching staff [17]. Similar to global education partners, Irish universities transitioned to remote online education during the initial lockdown period (March–June 2020), in addition to during the more stringent subsequent lockdown phases (October–December 2020 and January–April 2021), while alternating with a blended learning approach during periods of less stringent restriction (April–June 2021) [18]. The resultant fluctuations in delivery and related assessment models across the global university sector appeared to have further compounded and propagated the strain on student well-being [2,12]. In conjunction with this, several studies have examined the impact that this may have had on students’ academic performance [19,20]. In a study of 166 university students in the Netherlands during the stay-at-home period, a significant reduction in motivation was reported, while participants rated their online educational experience as ‘less satisfactory’ relative to traditional delivery modes [20]. This was reflected in less time allocated for university work/study during this period. At present, it appears that there is no targeted Irish research that has empirically investigated the impact of emergency remote delivery on student engagement within the university sector.

As an opportunistic cohort of interest, university students spend a prolonged period in university campus environments before they transition into the workplace [21]. The lifestyle habits that are established during students’ time in university are likely to be maintained for life; therefore, further exploration into student well-being is imperative within these settings [22]. Pragmatically, the findings of the current study will serve to provide invaluable insights for health promotion policy makers and practitioners within university settings, informing the identification of key areas of focus for future health promotion initiatives and an evidence-based rationale for the allocation of funding supports. 

Given the current paucity of empirical data pertaining specifically to the impact of COVID-19 on the health, well-being, and academic engagement of university students in Ireland, the purpose of the current study was to investigate how the COVID-19 pandemic and its associated societal restrictions impacted (i) health and well-being parameters, (ii) lifestyle behaviours, and (iii) academic engagement patterns of students within a multi-campus university in Ireland.

## 2. Materials and Methods

### 2.1. Study Design

A questionnaire-based cross-sectional design was adopted in this current study. A cohort of 2nd Year undergraduate students (*N* = 2752) within a multi-campus university setting in southern Ireland was purposively selected, as participants were requested to recall and report across four timepoints that coincided with key milestone stages of COVID-19, whilst concurrently transitioning to the university setting during the 18 months prior to data collection (April 2021), summarised in Table 1. 

The overarching methodological approach was informed by, and adapted from, previous work carried out within the host institution that investigated the habitual health and lifestyle behaviours of a cohort of 2267 students [15]. Ethical approval was sought and granted by the Host Institution’s Research Ethics Committee prior to data collection. 

### 2.2. Survey Implementation and Data Collection Procedure

The questionnaire was disseminated by email via the host university’s centralised email system during the Spring semester (April–June) of 2021. The email contained a direct hyperlink to the questionnaire instrument (Table 2), which was hosted on the ‘Qualtrics’ software platform (Version 4, Provo, Utah, USA). Participation was incentivised with entry to a draw to win one of three EUR 100 vouchers on completion of the questionnaire. Reminder e-mails were distributed after four weeks and seven weeks, resulting in the questionnaire remaining accessible for a total of 50 days to facilitate participants completing their end-of-year exams. Inclusion criteria included Year 2 full-time undergraduate students attending the host university at the time of data collection, aged ≥18 years. Participants were required to provide consent and confirm their age prior to progressing to complete the questionnaire instrument.

### 2.3. Questionnaire Instrument

The 68-item questionnaire instrument was part of a larger health and well-being study and was refined to focus on three distinct thematic sections including health-related behaviours (general health, physical activity (PA), body mass index (BMI), nutrition, alcohol, and sleep) mental well-being, and academic engagement (Table 2). Where relevant and appropriate, participants were asked to recall and report health behaviours/parameters across the four purposively defined timepoints (T0-T1-T2-T3), as defined in Table 1. To minimise participant burden, skip-logic was embedded within the questionnaire platform to ensure that they were only exposed to items of direct relevance. Further, a number of validated scales were incorporated, including the AUDIT-C and WHO-5 well-being index, which were scored according to their instruction manuals. A test of internal consistency was also carried out on both scales, using ‘Cronbach’s Alpha’ (Babor et al., 2001; Topp et al., 2015) Mental well-being was measured via the ‘World Health Organisation—Five well-being index’ [WHO-5] [24], which offers five questions relating to how a person has been feeling overall within the previous two weeks, before providing a well-being score for each individual. This index provides a well-being score for each individual, ranging from 0–100 with 0 representing the worst quality of life and 100 representing the best possible quality of life. 

Alcohol consumption patterns were measured and classified using the hazardous alcohol subscale (items 1–3) of the ‘Alcohol Use Disorder Identification Test’ [AUDIT-C] [23], which is a validated instrument endorsed by the World Health Organisation [25]. Similar to the work of Davoren et al. [26], the current study adapted the AUDIT-C scoring thresholds of five or more drinks for females and six or more drinks for males to constitute hazardous drinking.

### 2.4. Statistical Analysis

Data were exported from the Qualtrics platform for cleaning and analysis using IBM Statistical Package for the Social Sciences (SPSS-Version 27.0). Only fully completed responses were stored on the Qualtrics platform. Spoiled, implausible, outlier, and/or missing data points were reviewed and discussed by two members of the research team were deemed appropriate. To ensure consistency in analysis, all self-reported height and body mass data were mathematically converted to metric equivalents where required. BMI was calculated by the following formula [27]:BMI = body mass (kg)/height (m^2^)

Numerical data were summarised using means, standard deviations, and median/interquartile ranges, while all categorical data were summarised using relative frequencies. Health-related behaviours, well-being parameters, and academic engagement were (i) dichotomised by self-reported academic grade, prior to the COVID-19 pandemic (T0) (ii) analysed by gender. These were categorised as higher-grade category—‘60% or more’, or lower-grade category—‘59% or less’, based on the Irish grading system classifying 60% or more as a first- or second-class honours. Between-gender differences were explored using chi-squared tests for independence (categorical variables) and/or Mann–Whitney U tests for non-parametric numerical data as appropriate. Between-group gender analyses were primarily conducted between groups who identified as ‘male’ and ‘female’. Where relevant, descriptive analyses were also carried out for participants who selected ‘other’ or ‘prefer not to say’, although relative size of this group was low (n = 5). Paired sample t-tests were conducted to compare fruit and vegetable consumption across timepoints T0–T3. The alpha level of significance was set at *p* < 0.05 for all tests.

Binary logistic regression was used to identify significant predictors of participants’ self-reported body mass gain from the onset of COVID-19 to time of data collection. A dependant outcome dichotomised variable was derived as; 1 = increase in body mass, 0 = all other responses (including (i) no change in body habitus, (ii) that they had lost body mass, and/or (iii) didn’t know), based on two relevant questionnaire items (“Have you noticed a change in your body mass during COVID-19?”, and (if-applicable) “Do you feel you have? Gained weight/Lost weight?”). The independent variables selected, based on results from preliminary exploratory tests of association (chi-square tests), were gender, age group (18–20, 21–23, 24+), academic grade range (dichotomised), calculated BMI category, feelings about body mass, likert-type dietary statements, changes to sleep during lockdown, self-perceived BMI category, adherence to physical activity guidelines during lockdown (students’ semester one of 2nd year), habitual fruit and vegetable intake during lockdown (students’ semester one of 2nd year), perceptions of impact upon physical activity levels, and classified drinking status based on AUDIT-C ranking. For the logistic regression analysis, between the dependant variables of interest that were coded for the dichotomous outcome variable (1 = condition of interest, 0 = not of interest), there was no evidence of multi-collinearity amongst the selected independent variables. 

## 3. Results

### 3.1. Participant Demographics

Data were collected from a total of 268 participants, representing a response rate of 9.7%. The mean completion time of the questionnaire was 16.7 ± 5.3 min. Data from two participants were subsequently removed as they did not meet the inclusion criteria (did not agree to consent, under the age of 18), therefore data from 266 participants that fully completed the questionnaire were analysed. 

Of the total respondents, 28.6% identified as male (*n* = 76), 69.5% as female (*n* = 185), and 1.9% identified as ‘other’ or ‘preferred not to say’ (*n* = 5), with an age range from 18–56 years (median age 20 years, mean age 22.4 ± 5.8 years). Table 3 presents participant demographics stratified by gender.

### 3.2. Health-Related Behaviours

#### 3.2.1. General Health

At T3 (time of data collection), the majority of participants (72.6%, *n* = 193) rated their overall general health as either ‘*good*’ (56.8%, *n* = 151) or ‘*very good*’ (15.8%, *n* = 42), while gender was not significantly associated with general health rating (χ^2^ =3.808, *p* = 0.28). When participants were dichotomised by their self-reported academic performance, 71.4% of the higher-grade category were in this positive general health category (Table 4). Interestingly, 80.6% of participants in the lower-grade category rated their general health either ‘*good*’ or ‘*very good’.*


#### 3.2.2. Physical Activity

Prior to the onset of the COVID-19 pandemic (T0), a greater proportion of males (n = 60, 80%) reported that they were meeting the Irish PA guidelines (i.e., at least 150 min of moderate intensity activity per week or 75 min of vigorous intensity activity per week) in comparison to their female counterparts (n = 121, 66.1%). At each timepoint thereafter, females continued to report lower levels of PA than males, however the gap between genders reduced during T2 and T3, with males reported PA levels dropping by 17.4% between T1 and T2 (T1: 74.7% vs. 60.7%; T2: 57.3% vs. 55.7%; and T3: 66.7% vs. 61%). No significant associations were found between gender and sedentary levels (χ^2^ = 1.332, *p* > 0.514). Self-reported PA and sedentary levels displayed an inverse relationship from T0 through to T2 (see Figure 1), with PA levels across both genders decreasing by 55.6% (n = 148), while 62.7% (n = 165) of participants reported an increase in sedentary levels. 

#### 3.2.3. Body Mass Index

With regard to changes in body mass, 72.9% (n = 194) of participants perceived a change in their body mass from pre-COVID-19 (T0) to time of data collection (T3), with 78.4% (n = 152) reporting a gain in body mass. Participants’ mean self-reported body mass gain was 7.6 kg (range:1–30 kg). A relatively lower proportion (21.6%; n = 42) reported losing body mass, with a mean body mass loss of 8.4 kg (range: 1.36–26.8 kg).

Participants were asked how they felt about their current body mass (i.e., T3—time of data collection) compared to pre-COVID-19, with 35% reporting that they felt either ‘poor’ (24.1%), or ‘very poor’ (10.9%) in this regard. ‘Calculated’ versus ‘self-perceived’ BMI categories were examined based on those participants who reported both parameters correctly (n = 232), with participants who only reported one data point such as body mass, removed (Figure 2), revealing an apparent misconception regarding obesity levels (5.5% perceived prevalence vs. 11.8% calculated prevalence). A greater proportion of males perceived themselves as ‘underweight’ versus calculated scores (14.3% perceived vs. 8.6% calculated). Inversely, a greater proportion of females perceived themselves as ‘overweight’ (34.6% perceived vs. 31.5% calculated), but they under-reported the obese (2.6% vs. 12.3%) and underweight (2.5% vs. 4.9%) categories, respectively (Figure 2). There was a significant difference in accuracy of self-reporting BMI category between males and females (p = 0.002), see Figure 2. 

A regression model was utilised to investigate predictors of body mass gain. The model identified 61.3% (Nagelkerke R square) of the variance in predicted body mass gain and correctly classified 82.6% of cases. The percentage of cases that were correctly predicted by the model (i.e., the sensitivity) was 86.7%. The specificity of the model that correctly predicted participants that reported either (i) no change in body habitus, (ii) that they had lost body mass, and/or (iii) didn’t know, was 76.7%. The positive predictive value, the percentage of correctly predicted cases with the observed characteristic compared to the total number of cases predicted as having gained body mass was 84.3%. The negative predictive value, which is the percentage of correctly predicted cases without the observed characteristic compared to the total number of cases predicted as not having gained body mass, was 80.0%.

The model identified four significant independent variables of the predictors of body mass gain; namely those (i) aged > 24 years, (ii) feeling ‘poor’ or ‘very poor’ about body mass, (iii) perceiving their BMI as overweight/obese, and (iv) reporting more sleep. There was no evidence of multi-collinearity from either (i) bivariate correlations between independent variables, or (ii) collinearity diagnostics (tolerance >0.10, and Variance Inflation Factor (VIF) between 1.1–1.9). An inspection of standardised residual values revealed that there were ten outliers, (Std. Residual range from −2.478–2.139), which were kept in the dataset. Participants were less likely to have gained body mass if they ‘agree,’ ‘neither agree or disagree’ or ‘disagree/strongly disagree’ compared to those who ‘strongly agreed’ with the dietary statement ‘I sometimes eat because I am bored’. Interestingly, those who reported ‘poor’, or ‘very poor’ sleep quality were less likely to gain body mass (Table 5). 

The data also revealed that gender was not an independent variable that predicted the likelihood of body mass gain. The highest predictor of likelihood of body mass gain related to participants aged >24 years, who were 8.1 times more likely to have reported body mass gain in comparison to those in the 18–20 and 21–23 age categories (*p* = 0.12). Those who reported feeling ‘poor/very poor’ about their body mass were 4.9 times more likely to report body mass gain than those who reported ‘good/very good’ (*p* = 0.00). In relation to participants who perceived themselves to be overweight/obese, they were 5.71 times more likely to report body mass gain versus those who perceived themselves as normal weight/underweight (*p* = 0.03). Sleep quality and sleep duration were both predictors of body mass gain, with those who reported more sleep presenting as 6.4 times more likely to report body mass gain (*p* = 0.01), and those who reported ‘poor/very poor’ sleep quality presenting as less likely to have reported body mass gain in comparison to those whose sleep was ‘good/very good’ (OR = 0.23, *p* = 0.08). Similarly, those who answered, ‘strongly agree’ to the dietary statement ‘*I sometimes eat because I am bored*’, were those more likely to have gained body mass versus those who answered ‘agree’ (OR = 0.22) to ‘disagree/strongly disagree’ (OR = 0.02, *p* = 0.00).

#### 3.2.4. Nutrition & Alcohol

Daily fruit and vegetable servings consumption increased from 3.9 mean daily servings at T0, to 4.0 mean daily servings at T1, to 4.2 mean daily servings T2; but this increase was not reported as significant (*p* > 0.05) via a paired sample t-test. However, there was a significant difference between the mean consumption of daily fruit and vegetable servings from T0 (pre-COVID) to time of data collection T3 (April–June 2021) (3.9–4.3 mean daily servings, *p* = 0.014). A greater proportion of participants who increased their overall fruit and vegetable consumption ‘*strongly agreed*’ (12.6%) or ‘*agreed*’ (24.4%) that they ‘*were more aware of their calorie consumption*’ at T3 (χ^2^ = 68.266, *p* = 0.044). Furthermore, there was a significant association between those who ‘*strongly agreed*’ (13.5%) or ‘*agreed*’ (23.0%) that they were ‘*more likely to have set mealtimes*’ and those who reported an increase in fruit and vegetables at T3 (χ^2^ = 62.243, *p* = 0.014).

At time of data collection (T3), 82.3% (*n* = 221) of participants reported drinking alcohol habitually, comprising 80.3% of males, (*n* = 61), 84.3% females, (*n* = 29), and 80.0% other, (*n* = 4). The AUDIT-C scale exhibited similar reliability to the previous work of Bickerdike et al. (2019), with a Cronbach’s Alpha of 0.62. Worryingly, based on the AUDIT-C threshold scores, 63.9% (*n* = 39) of male and 42.2% (*n* = 65) of female drinkers were categorised as ‘hazardous’ drinkers [26]. Further, significant differences were revealed between these male and female AUDIT-C, with more males categorised as hazardous drinkers than females (*p* = 0.004). Regarding the initial impact of the stay-at-home period (T1), 66.5% (*n* = 147) of participants reported changes to their drinking habits with increases in either ‘frequency’, or ‘frequency and volume’, accounting for 41.5% (*n* = 61) of this cohort. Interestingly, 52.4% (*n* = 77) reported a decrease in such drinking habits. (Figure 3). Alcohol consumption habits at T2 were analysed similar to T1, and there was no significant association between gender and reporting a change in alcohol consumption at T2 (55.7% males reported a change vs. 47.4% of females), χ^2^ = 1.209, *p* = 0.272. 

#### 3.2.5. Sleep 

In terms of self-reported habitual sleep duration during T3, 25.9% of participants reported eight hours of sleep per night on weekdays, which increased to 32.3% on weekends. There was no significant association between gender and sleep duration on weeknights (χ^2^ = 16.82, *p* > 0.156). However, there was a statistically significant difference between gender and attaining eight hours of sleep per night at the weekend (χ^2^ = 26.97, *p* = 0.008) with 8.6% of males vs. 22.9% of females reporting they obtained this recommended duration. In total, 41.7% (*n* = 111) of participants reported that they got more hours of sleep during the pandemic (T1, T2 & T3) relative to the period prior to its arrival. Contrary to participants’ reported sleep quantity during COVID-19, results revealed that sleep quality scores decreased during these timepoints with 35% reporting ‘*poor*’ or ‘*very poor*’ sleep quality during T3, compared to 16.6% reporting these ratings during T0. Figure 4 presents the percentage of participants who rated their sleep quality as either ‘Very Good’ or ‘Very Poor’, at each of the four COVID-19 related timepoints.

### 3.3. Mental Well-Being

Findings revealed that 34.6% (*n* = 92) of participants rated their mental well-being at T3 as ‘good’, 28.6% (*n* = 76) as ‘neither good nor poor’, 19.2% (*n* = 51) as ‘poor’, and 10.2% (*n* = 27) as ‘very poor’. Worryingly, only 7.5% (*n* = 20) rated their mental well-being as ‘very good’. Notably, 38% (*n* = 101) reported receiving some form of treatment or support for their mental well-being, either at the time of data collection (T3—13.9%, *n* = 37), or at some time in their life (24.1%, *n* = 64). Of those participants that reported receiving such treatment or support, 18% reported that this was via a friend/family member, 13.9% stated that they had previously spoken to a lecturer/member of staff, 11.7% reported that they had attended their GP, 18% stated that they previously attended a counsellor, 12.4% reported attending a psychiatrist, while 10.2% stated that they had previously received in-patient psychiatric care. Gender did not discriminate between self-rated mental well-being (χ^2^ = 3.70, *p* > 0.447).

The WHO-5 well-being index exhibited a high level of reliability, as indicated by Cronbach’s alpha (0.86). Regarding the WHO-5 well-being index [24] (range: 4–100), only one participant (male) achieved the maximum of 100 points, while the median male score was 52 (IQR ± 28: the data was determined to be not normally distributed). Gender differences were evident, however, with 66% of males scoring 50 or more in the WHO-5 scale, relative to 46% of females. Those who identified as ‘other’ exhibited lower mean and median scores than both males and females (Mean: Male—54/Female—46.75/Other—37.6; Median: Male—60/Female—48/Other—28). To further explore differences between the gender groups, a Kruskal–Wallis test was applied, revealing a significant difference (*p* < 0.006) between the three groups, with post hoc tests confirming significant differences between the males vs. females (but not males vs. other, females vs. other). A Mann–Whitney U test was conducted to determine the gender-differences in WHO-5 scores, revealing statistically significant differences between males and females, U = 5295, z = −3.028, *p* = 0.002. The options of ‘all of the time’ or ‘most of the time’ were the most frequent answers by participants in response to the statements within this well-being index (Figure 5). 

### 3.4. Educational Experience & Academic Engagement

A sub-set of specific aspects that constitute the University experience were rated on a 5-point Likert scale, including (1) ‘*communication with lecturers*’, (2) ‘*using online learning management*’, (3) ‘*communication with classmates*’, (4) ‘*accessing supports*’, and (5) ‘*attending online lectures/webinars*’. Figure 6 provides an outline of the effects of the COVID-19 pandemic on participants reported academic engagement at each timepoint. A notable decline in the number of participants who rated ‘attending online lectures/webinars’ as ‘good’ is particularly evident here. These findings also demonstrate a slight increase in the number of participants who rated (5) as ‘very poor’ during the same timepoints (T1—7.1%; T2—5.3%; T3—8.6%). 

When reporting on their experience of university across all four timepoints (T0–T3), there was an overall decline in participants feeling ‘very good’ or ‘good’ (Figure 7). Inversely, students who rated their university experience as ‘very poor’ and ‘poor’ at T0 increased their rating incrementally across the timepoints (Figure 7). This demonstrates some of the negative impacts that the COVID-19 pandemic had on students. 

## 4. Discussion

The current study serves to provide an insight into the health-related behaviours, well-being, and academic engagement of a cohort of Irish university students (*n* = 266) prior to, and during, the global COVID-19 pandemic. Considering the devasting impact of COVID-19 on population health and wellbeing worldwide, and despite living through this difficult period, the majority of students, 72.6%, rated their general health ‘*good*’ or ‘*very good*’ at the time of data collection (T3). Despite the maintenance of students’ general health*,* these figures are substantially below age group-related norms reported in similar Irish research, where 93% of both males and females aged 15–24 reported their general health as ‘*good*’ or ‘*very good*’ [14]. Similar to the findings of Bickerdike et al. [15], no student in the current study rated their ‘general health’ as ‘*very poor*’, despite living and studying through the COVID-19 pandemic. The overall self-reported optimistic general health rating was not seen globally, and contrasts with the findings in Japan by Tahara et al. [28], who used the General Health Questionnaire-12 instrument (GHQ-12) to measure the self-reported health of university students and found that 70.9% of participants reported poor mental well-being, which was higher than before COVID-19. Further, less communication with friends was reported by this cohort, whilst leisure and new activities were correlated to students with higher scores. A similar study conducted by Rafal et al. [29] also used the same GHQ-12 instrument and found that feelings of loneliness and poor perception of family environment resulted in poor psychological scores [29]. Therefore, cultural differences, and the severity of stay-at-home orders that were in place in different countries across the globe, may have contributed to the difference between general health scores. 

Several studies have reported a stronger correlation between males engaging in weekly physical activity levels than their female counterparts [15,30,31,32,33]. The current study revealed a similar trend, with 66.7% of males reporting engagement in PA levels weekly, in comparison to only 61% of females. However, an overall reduction in PA levels was reported across both genders across the four timepoints, while an increasing trend in sedentary time was evident. Interestingly, although this overall decrease was observed in the PA levels of ‘both’ genders, males displayed a greater reduction in respective PA levels relative to that of their female counterparts, and particularly between T1 (74.7%) and T2 (57.3%). These findings may be a direct result of the significant increase (*p* = 0.004) in hazardous drinking habits displayed by males during the COVID-19 pandemic. López-Valenciano et al. [33], conducted a systematic review examining the impact that COVID-19 had on students PA levels and reported a significant reduction in the physical activity levels of students in nine of the ten studies reviewed. 

In terms of body habitus, 53.6% of participants classified themselves within the ‘normal weight’ BMI category, with 28.7% and 11.8% classified as ‘overweight’ and ‘obese’, respectively. From the cohort that elected to report their self-perceived BMI category, whilst also reporting their height and body mass (*n* = 232), it was revealed that both genders underestimated their BMI category. Regarding males, 22.4% perceived themselves as ‘overweight’ with 31.4% calculating within this category, and 1.8% perceiving their BMI as ‘obese’ with a calculated result of 10.7%. Inversely, more females perceived themselves as overweight (34.6% vs. 31.5% calculated) and similarly to males, they under-reported the obese category, with only 2.6% perceiving themselves as being in this BMI category versus the 12.3% calculated scores. It is apparent that university student cohorts have a lack of understanding of the classification of BMI category as similar findings were reported by Bickerdike et al. [15], whereby a greater proportion of females classified themselves as ‘overweight’ but again underreported the ‘obese’ category, relative to their ‘calculated scores’ (30.7% vs. 23.0% and 4.0% vs. 11.3%, respectively). This is critically valuable information for policy makers and, of greater concern is that fact that as both university cohort-based studies relied on self-reported data, the true extent of the degrees of overweight and obesity may be underreported. 

A substantial proportion of participants (72.9%) reported a change in body mass from the onset of COVID-19 (T1) to time of data collection (T3), with five times as many participants indicating increases in body mass versus a loss in body mass. In a similar study conducted by Palmer et al. [11], data were collected from 827 University students in Germany who completed an online questionnaire to examine their changes in lifestyle, diet, and body mass during the first COVID-19 stay-at-home period (March–May 2020—T1 in the current study) and found almost half of the participants (46.4%) reported a change in body mass, or which 27.5% reporting a gain while 21.9% perceived that they lost body mass. A subsequent regression analysis by Palmer et al. [11], revealed that increased consumption of pasta, meat, sweets, cakes, and savoury snacks during the period of investigation were associated with body mass gain. Jalal et al. [34] conducted interviews with 628 University students in Saudi Arabia and reported that 32% of participants gained body mass from March–June 2020 during the COVID-19 pandemic (T1 in the current study). Interestingly, the prevalence of increased body mass throughout COVID-19 was highest in the current study with 78.4% of students who reported a change in body stating that this change related to a gain. The two previous studies [11,35] were conducted within the first six months of the onset of COVID-19 and the results reported heretofore may have been larger if carried out at the same period of the current study. With specific regard to eating behaviours in the current study, it was evident that negative habits such as eating when bored (as opposed to when hungry), feeling ‘*poor’ or ‘very poor*’ about body mass, perceiving their BMI as overweight/obese, reporting more sleep, and having a poorer sleep quality, were strong predictors of body mass gain. This may have been due to the lack of social connectivity during the stay-at-home periods, which resulted in an increased caloric intake. Those tasked with this responsibility should strive to create interventions that (i) include educating students on intentional healthful dietary habits to encourage the targeting of the recommended dietary intakes of macro- and micro-nutrients and (ii) focus on increasing daily physical activity levels to combat the increase in excess body mass gain to combat the negative behaviours incurred during COVID-19. 

Achieving the recommended daily intake of fruit and vegetable servings per day (five-seven servings) has many health benefits [35]. In some instances, the COVID-19 pandemic has increased consumption of fruit and vegetable intake, but this is not the case globally. From a localised perspective, the current study revealed that there were positive increases in participants’ fruit and vegetable intake. Fruit and vegetable consumption remained suboptimal despite a significant increase (*p* = 0.014) from pre-COVID-19 (September-December 2019—3.9 servings) to the time of data collection (April–June 2021—4.3 servings), with the median intake of four servings per day across the timepoints. This self-reported fruit and vegetable intake is higher than the figures reported by Bickerdike et al. [15] among a larger sample (*n* = 2267) of university students in which the median fruit and vegetable intake was found to be three servings per day. In contrast to these findings, a nationwide survey-based study conducted in China during the COVID-19 pandemic that examined changes in dietary patterns among a youth cohort aged 15–28 years identified a significant decrease in the frequency of fruit and vegetable intake [9]. However, a US university-based cross-sectional study (N = 291) reported no significant change in fruit and vegetable consumption, the frequency (pre-COVID) and during the first stay-at-home period (March–May 2020) remained at median vegetable consumption three portions per day and two portions of fruit per day [12]. With regard to the current study, the positive trend certainly needs improving, while the findings of the related studies above [9,12,15] highlight the need for greater education around the importance of adequate fruit and vegetables consumption amongst this cohort as well as improving the provision of dietary choices within a university setting which could be enhanced by policy holders committing to ensuring food offerings have increased fruit and vegetable options. 

Despite the stay-at-home measures that were in place due to COVID-19, high levels of hazardous drinking habits were reported by both males and females in the current study, with 63.9% of males consuming ‘six or more drinks’ and 42.2% of females consuming ‘five or more drinks’ on a typical night out despite the lack of social settings (i.e., bars, nightclubs, etc.) at this time. Relating back to the work of Bickerdike et al. [15] in the same university setting as the current study, hazardous drinking prevalence for males and females was 54.7% and 54.1%, respectively, this indicating a decrease in hazardous drinking for females but a worrying increase for their male counterparts. A study conducted by Davoren et al. [26], amongst 2275 undergraduate students in a different Irish university setting also revealed high levels of hazardous drinking with 65.2% of males and 67.3% of females reporting drinking levels that exceeded six drinks per males and five for females in one night. The decrease in hazardous drinking levels with regard to females between these two studies may be due to the impact of the stay-at-home measures and organised student nights during this time; therefore, this positive change in hazardous drinking levels may not remain once stay-at-home orders are abolished. However, males’ levels of hazardous drinking have fluctuated, and remain a cause for concern. This may be due to the decrease in physical activity engagement, which was evident in the current study, allowing for more time for recreational drinking and/or commitment to sporting teams that require monitoring, or abstinence of drinking ceasing during this period. As this is self-reported data, and as the standard public houses measures of alcoholic units are removed, the single measure volumes may have in fact been under-reported, resulting in even larger volumes of alcohol being consumed. Interestingly, the hazardous drinking levels reported by the male cohort in the current study are similar to the most recent Irish statistics [14] (63.9% of males versus 44% aged 15–24 years). Contrasting trends were seen in the female cohort (42.2% in the current study versus 16% in the same previous Irish research) [14]. On the contrary, a US study amongst 439 1st Year students, that compared data from pre-COVID (October 2019–February 2020—T0 in the current study) to during the pandemic (June–July 2020—T1 in the current study), revealed that binge drinking decreased from 35.5% to 24.6% [36]. 

Regarding general alcohol consumption habits since the onset of the COVID-19 pandemic, 66.5% of participants in the current study reported changes in alcohol consumption patterns. Other similar studies conducted during COVID-19 reported that 96.5% of university students changed their drinking habits, and, similar to the current study, there was not a significance between groups that decreased or increased their drinking habits [36]. Positive trends have been in Belgium, where 68% of university students surveyed (*n* = 1951) about their alcohol consumption during April–May 2020 (T1 in the current study) reporting a reduction in their alcohol consumption substantially by more than 12 units per week since the onset of COVID-19 [37]. As alcohol contains a high calorie content, and low nutritive value [35], and is associated with negative health implications such as disturbed sleep, sleep disturbances, and hinders the absorbance of certain nutrients such as folate, vitamin A and B12, magnesium and calcium to name but a few, [38,39], there should be a strong focus within the university setting on encouraging sensible alcohol consumption within the recommended limits for genders. 

In the current study, 3.8% of participants rated their sleep quality as ‘*very poor*’ during T0, with this figure increasing to 12.4% at T3, representing a concerning decrease in sleep quality over the COVID-19-related timepoints. Similarly, 50.8% of participants rated their sleep quality as ‘*good*’ during T0, however, this decreased to 33.1% by T3, outlining a major reduction in participants’ sleep quality. This may be because of the added pressure endured by students as end of semester university exams were ongoing, during the time of data collection. Further analysis across the 4 timepoints revealed a decrease (4.2%) in participants who rated their sleep quality as ‘*very good*’ between T0 and T1. Interestingly, however, 41.7% (*n* = 111) of participants reported attaining ‘more sleep’ per night during the lockdown periods, thereby suggesting sleep volume does not equate to sleep quality, within this data set. Martínez-de-Quel et al. [40] investigated the sleep quality of students (*n* = 693) both before and during the COVID-19 pandemic and found that the confinement due to the COVID-19 pandemic resulted in a significant decline in the quality of university students’ sleep, thus substantiating the findings observed in the current study. 

A plethora of research in recent years has demonstrated the negative effects of sub-optimal sleep patterns on academic performance, such as poor brain function and reduced memory functioning [41,42,43,44,45,46]. Research has also highlighted the critical role sleep plays in a person’s productivity and performance levels [47]. In the aforementioned logistic regression model applied in the current study to identify factors influencing body mass gain across the four stated timepoints, a key finding related to the association between increased sleep duration and body mass gain. However, the model also found that those who were less likely to gain body mass had reported poor or very poor sleep quality. These findings contrast previous reports that both sleep quality and sleep quantity aid weight loss [48,49,50]. Interestingly, 50.8% of participants reported their sleep quality as ‘*good*’ before the onset of the COVID-19 pandemic (T0), while this decreased to 33.1% at the time in which they were completing the questionnaire (T3). This finding may come as a direct result of the added stress to students at the time of data collection due to university exams. These findings suggest the requirement for a university-based intervention aimed at improving students sleep habits. 

Within the current study, an array of findings relating to students’ ratings of their mental well-being was reported over the four timepoints. During T3, 10.2% (*n* = 27) of participants rated their mental well-being as ‘*very poor’,* with only 7.5% (*n* = 20) rating their mental well-being as ‘*very good’.* An extensive research study by Ammar et al. [8], comprising 35 research organisations across four continents, revealed similar findings relating to the negative effects on mental well-being and the emotional state of its general population participants (*n* = 1047), due to the COVID-19 pandemic. The current study also unearthed stark findings related to student mental well-being and related use of support services, with 10.2% of those who previously admitted to receiving treatment or support for their mental well-being stating that they had previously received inpatient psychiatric care. To date, there is a dearth of research examining university student cohorts who have sought in-patient care for their mental well-being. Pedrelli et al. [51] outlined the importance of identifying student’s mental well-being issues and subsequent provision of accessible and appropriate treatments to support this concerning issue. 

Evidence of positive academic engagement in university students at the onset of the COVID-19 pandemic (T1) was also identified. However, the initial positive response evidentially diminished over time, as the number of participants who rated their experiences of online learning as ‘*Poor*’ or ‘*Very Poor*’ began to increase drastically during T3 (T1 = 16.4%; T3 = 25.2%). Similar findings were observed in a study conducted in the Netherlands (*n* = 166 university students), where it was reported that participants found their online educational experience less satisfactory than traditional in-person lectures [20]. To date, there is a dearth of research exploring university academic engagement levels after the successive COVID-19 enforced lockdown phases. 

There are several practical implications of the current study. The use of validated scales and derivation from other Irish studies facilitated relevant comparisons of data across a range of thematic areas [15,26] particularly with regard to the period pre-pandemic. The breadth of data collected, and associated findings will be of interest to practitioners and policy makers whose remit extends to the health, well-being, and/or academic engagement of students. 

Although the current study appeared to be the first of its kind in an Irish context, there are some limitations that should be noted. This was a cross-sectional study; therefore, findings are limited to relevance and context of the time of data collection and preclude any inferences in terms of causality. That being said, this study has produced a critical overview of the health and well-being of a cohort of university students and provides a valuable insight into the impact of COVID-19 on students’ progress through university across this period. Inevitably, the use of self-reported data may have led to under- or over-reporting in some cases. Future studies may benefit from conducting anthropometric measures to determine greater accuracy in this aspect of the research design. The relatively small sample size in the current study may also be considered a limitation, in addition to the fact that there was a larger response rate from females than males.

## 5. Conclusions

As the COVID-19 pandemic is ever-evolving and uncertain, this is a time-critical area for research, with several related studies already conducted in other countries, yet none in an Irish university setting. These studies have identified the major implications experienced by the university student population as a direct effect of the COVID-19 pandemic [4,9,11,13]. Findings in the current study revealed several sub-optimal and concerning findings. Body habitus status remains a cause for concern with over two thirds of participants classified in the overweight and obese categories, Predictors of body mass increases included age (>24 years), feeling ‘*poor/very poor*’ about body mass, perceiving BMI as overweight/obese, strongly agreeing with the dietary statement ‘*I sometimes eat because I am bored’,* getting more sleep during stay-at-home period, and having very poor sleep quality. These should be considered by policy makers with regard to how healthy lifestyle habits may be facilitated amongst university student cohorts to combat obesity. Suboptimal trends in fruit and vegetable intake were revealed across the 4 pandemic timepoints and need continuous improvement. Throughout COVID-19, hazardous drinking rates were still worryingly high amongst participants and therefore efforts to reduce these levels should be a focus for future health and well-being initiatives. Participants who were classified in the higher risk drinking category were also more likely to have changed their drinking habits during the stay-at-home period with changes in their frequency, volume, or both. 41.7% (*n* = 111) of participants reported ‘more sleep’ during the lockdown periods, sleep quality reported to be rather low. Therefore, it is imperative for the university setting to focus on encouraging good sleep hygiene. Whilst of significant concern is that fact that results also outlined stark findings in relation to the gender differences in mental well-being, with males obtaining higher scores of positive well-being in the WHO-5 scale than those of females (Males = 66%; Females = 46%). Of the seven questions that enquired about the four timepoints within this study, academic engagement throughout the pandemic provoked the greatest drop-in satisfaction rate. Interestingly, nearly 11% (10.9%; *n* = 29) of participants reported spending more than seven hours on their phone on an average weekday, which could be a contributing factor to the aforementioned decrease in students’ academic engagement or indeed the increase in poor student mental well-being. Overall, these data serve as an essential resource in contributing to the development of university-based initiatives to support the health and well-being of contemporary student cohorts.

## Figures and Tables

**Figure 1 ijerph-19-16096-f001:**
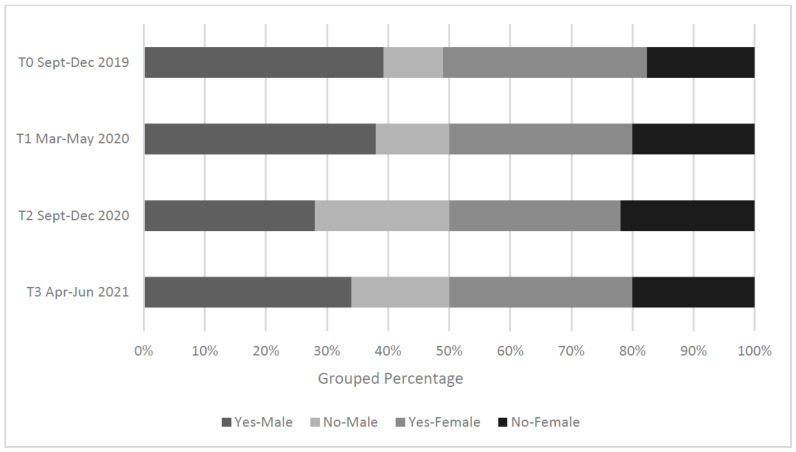
Percentage of male (n = 75) vs. female (n = 183) participants who reported meeting the government suggested guidelines for PA levels during the four COVID-19 related timepoints as outlined in this study.

**Figure 2 ijerph-19-16096-f002:**
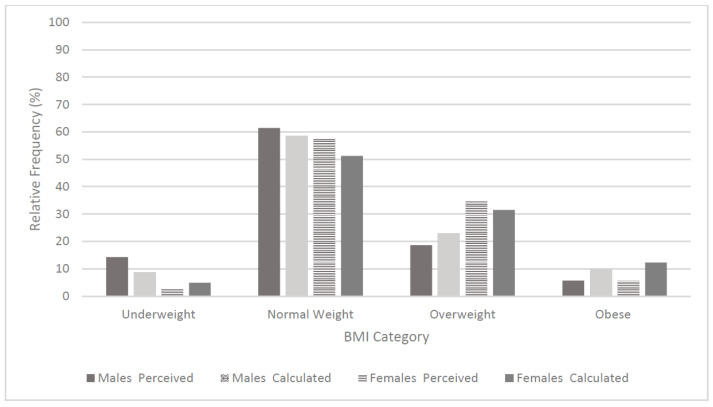
Received versus calculated BMI category by gender at T3.

**Figure 3 ijerph-19-16096-f003:**
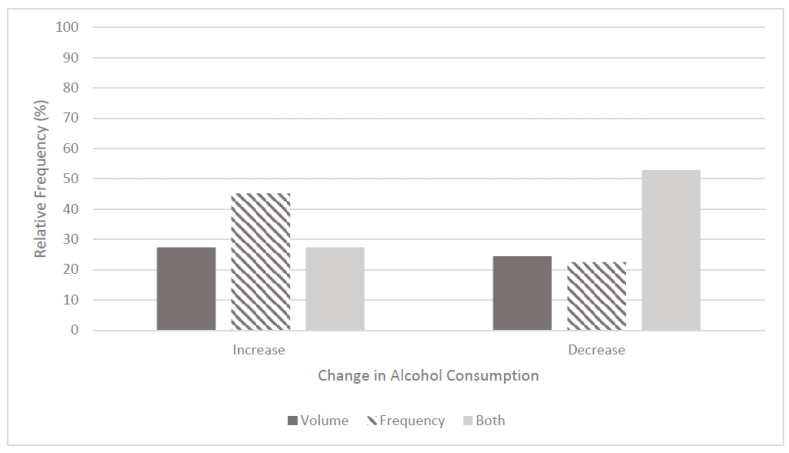
Self-reported changes in participants’ volume and/or frequency of alcohol consumption T1 (March–May 2020).

**Figure 4 ijerph-19-16096-f004:**
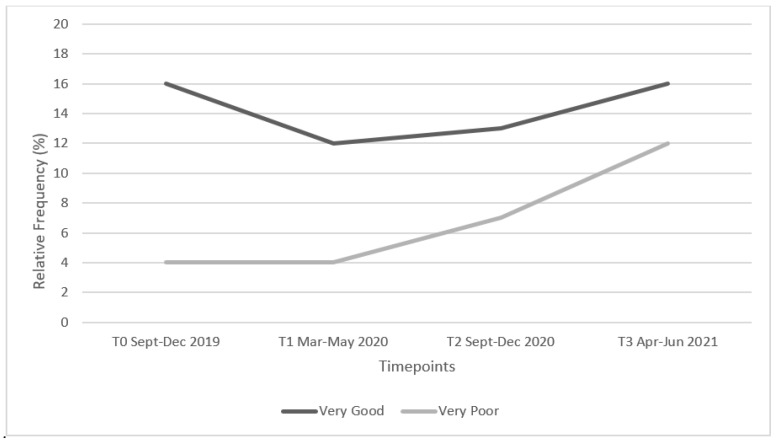
Percentage of participants who rated their sleep quality as ‘*Very Good*’ or ‘*Very Poor*’ at the four timepoints during COVID-19.

**Figure 5 ijerph-19-16096-f005:**
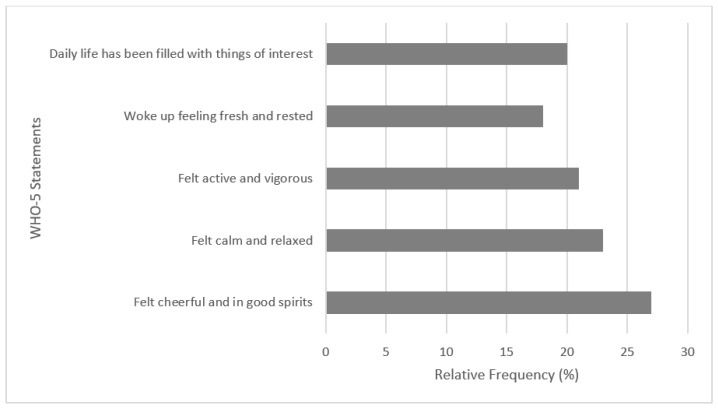
Percentage of participants who described feeling the above statements, ‘all of the time’ or ‘most of the time’, within the last 2-weeks.

**Figure 6 ijerph-19-16096-f006:**
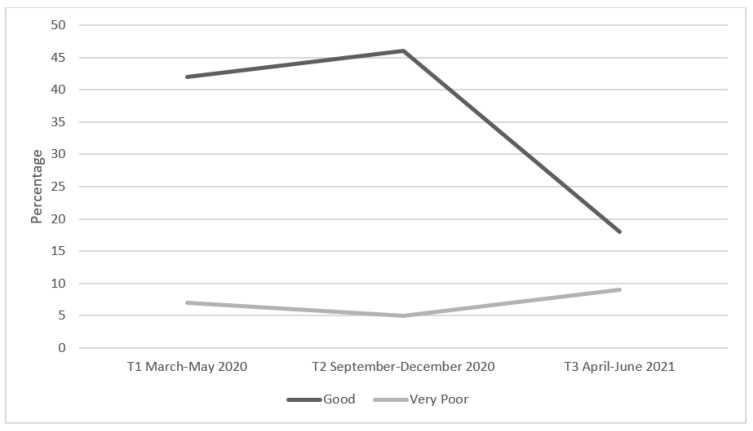
Percentage of participants who rated (5) ‘attending online lectures/webinars’ as ‘*Good*’ or ‘*Very Poor*’.

**Figure 7 ijerph-19-16096-f007:**
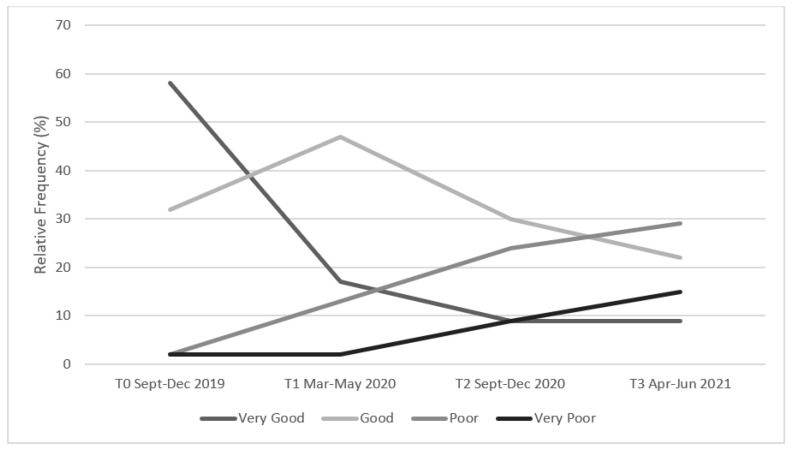
Percentage of participants who rated their university experience as ‘*Very Good*’, ‘*Good*’, ‘*Poor*’ and ‘*Very Poor*’ at four timepoints during COVID-19.

**Table 1 ijerph-19-16096-t001:** Key milestone stages of reporting and participants academic year outline.

Timepoint	COVID-19 Timeline	Corresponding Months & Cohort Academic Year	COVID-19 Status
Baseline (T0)	Prior to the onset of COVID-19	September–December 2019, Year 1 Semester 1	No restrictions in place
Timepoint 1 (T1)	The onset of COVID-19	March–May 2020, Year 1 Semester 2	Emergency transition to remote delivery
Timepoint (T2)	During COVID-19	September–December 2020, Year 2 Semester 2	Blended model with second transition to remote delivery in October
Timepoint (T3)	The time of data collection	April–June 2021	Remote or blended model, dependent on the academic Dept./discipline

**Table 2 ijerph-19-16096-t002:** Questionnaire thematic areas (including validated scales and refined items from previous research for the current study).

Thematic Area	Questionnaire Item	Adapted from (Where Relevant)	Questionnaire Item Structure
Demographic information	Gender/Age (self-reported in years)		‘*Male’, ‘Female’, ‘Prefer not to say’, ‘Other’/*Continuous
	Academic grade achieved in semester 1 of year 2 (T2)	[15]	‘*Less than 40%*’, ‘*40–59%’, ‘60–69%’, ‘70% or above*’, ‘*Do not know*’, ‘*I would rather not provide this information*’ and ‘*Other*’
**Health-related behaviours**			
General Health	Overall general health rating (5-point Likert Scale)		‘*Very good’, ‘Good’, ‘Neither good nor poor’, ‘Poor’, ‘Very poor*’
Physical Activity	Impact of COVID-19 on PA levels		‘*Yes’, ‘No*’
	Impact of COVID-19 on PA/sedentary levels, frequency and duration during (T0–T3)		‘*Increased’, ‘Decreased’, ‘Stayed the same*’
BMI	Self-reported body mass & height measurements/Perceived BMI category		Students could answer in imperial or metric values. Options available: ‘*underweight’, ‘normal weight’, ‘overweight’ or ‘obese*’
	Change in body mass/Gained or lost body mass/Feelings towards body mass		Have you noticed a change in your body mass during COVID-19? ‘*yes’, ‘no- I have stayed the same’, ‘do not know’, ‘gained body mass’, ‘lost body mass’, ‘stayed the same’/* ‘*Very good’, ‘Good’, ‘Neither good nor poor’, ‘Poor’, ‘Very poor*’
Nutrition	Eleven newly devised lockdown dietary habits statements such as: ‘*I have more time to prepare meals’, ‘I have more set mealtimes’, ‘I don’t eat as many takeaways’, ‘I sometimes eat because I am bored*’		Likert scale reporting level of agreement: ‘*Strongly Agree’, ‘Agree’, ‘Neither Agree nor Disagree’, ‘Disagree’, ‘Strongly Disagree’, ‘Not Applicable*’
	Daily fruit & veg intake across timepoints		*Self-reported sliding scale to indicate habitual daily portions (range 0–10) across timepoints T0* *–* *T3*
Alcohol	AUDIT-C scale	[23]	‘*How often do you have a drink containing alcohol?’ ‘How many units of alcohol do you drink on a typical day when you are drinking?’ ‘How often have you had 6 or more units if female, or 8 or more if male, on a single occasion in the last year?*’ Scores are calculated using a scoring scale with answers equivalent to a number. Calculated score thresholds of 5 or more for females and 6 or more for males to constitute hazardous drinking (Davoren et al., 2015).
	Impact of COVID-19 on drinking habits/did habits change during level-5 lockdowns		‘*Yes’, ‘No*’ If yes: Did your drinking volumes and/or frequency Nominal: ‘*Increased’, ‘Decreased*’
Sleep	Sleep quality across (T0–T3) (5-point Likert Scale)		‘*Very good’, ‘Good’, ‘Neither good nor poor’, ‘Poor’, ‘Very poor’, ‘I would rather not say*’
	Sleep duration at the time of data collection	[15]	‘*Less than 4 hours’, ‘4 h’, ‘5 h’, ‘6 h’, ‘7 h’, ‘8 h’, ‘9 h or more*’
	Changes in sleep duration across (T0–T3)		‘*More sleep’, ‘Less sleep’, ‘Stayed the same’, ‘I don’t know*’
**Mental Well-being**			
Perceived mental well-being/WHO-5	Personal rating of mental well-being (5-point Likert Scale)	[15]	‘*Very good’, ‘Good’, ‘Neither good nor poor’, ‘Poor’, ‘Very poor’, ‘I would rather not say*’
	Current and/or past receipt of mental well-being supports		‘*Yes’, ‘No’, ‘I would rather not say*’
	Source, type, and time period of support received		‘*Ongoing’, ‘Within the last month’, ‘Within the last 6 months’, ‘Within the last year’, ‘More than 1 year ago*’
	WHO-5 well-being index	[24]	
**Academic Engagement**			
Educational experience & academic engagement	Experience when entering university, ‘*socially’, ‘academically’/student’s ‘sense of belonging’ perceived university experience across timepoints: (T0-T3)/Rating of remote learning experience during applicable timepoints (T1, T2, T3)/‘Communication with lecturers’, ‘Using online learning management’, ‘Communication with classmates’, ‘Accessing supports’, ‘Attending online lectures/webinars*’		‘*Very good’, ‘Good’, ‘Neither good nor poor’, ‘Poor’, ‘Very poor’, ‘Not Applicable*’
	Daily technology use—weekday (Mon-Fri), weekend (Sat-Sun)		‘*Less than 1 h’, ‘1–3 h’, ‘3–5 h’, ‘5–7 h’, ‘7–9 h’, ‘9–11 h’, ‘More than 11 h*’

**Table 3 ijerph-19-16096-t003:** Participant demographics by gender.

		Total		Males		Females		Other *	
		*n*	%	*n*	%	*n*	%	*n*	%
Age (years)	18–20	149	56.0	42	55.3	104	56.2	3	60.0
	21–23	70	26.3	21	27.6	48	25.9	1	20.0
	24+	47	17.7	13	17.1	33	17.8	1	20.0
	Total	266		76		185		5	
Self-reported academic grade category Year 1 Semester 1 (pre-COVID)	Less than 40%	3	1.1	1	1.3	2	1.1	0	0.0
	40–59%	33	12.4	15	19.7	17	9.2	1	20.0
	60–69%	109	41.0	29	38.2	78	42.2	2	40.0
	70% or above	90	33.8	22	29.0	67	36.2	1	20.0
	Do not know	25	9.4	7	9.2	18	9.7	0	0.0
	I would rather not provide this information	6	2.3	2	2.6	3	1.6	1	20.0
	Total	266		76		185		5	

* Other is a combination of categories ‘*Prefer not to say*’ and ‘*Other’.*

**Table 4 ijerph-19-16096-t004:** Participants self-reported health and lifestyle variables stratified by gender and by dichotomized grade category.

Lifestyle Factor	Category	Total		Males		Female		Other	
		*n*	%	*n*	%	*n*	%	*n*	%
In general, would you say your health is?	Good or very good	193	72.6	58	76.3	131	70.8	4	80.0
	*Higher-grade category*	142	71.4	37	72.5	102	70.3	3	100.0
	*Lower-grade category*	29	80.6	15	93.8	14	73.7	0	0.0
BMI Category	Overweight or obese	96	40.5	23	32.9	71	43.8	2	40.0
	*Higher-grade category*	75	42.9	17	36.9	57	45.3	1	33.3
	*Lower-grade category*	11	32.3	3	20.0	7	38.9	1	100.0
Have you noticed a change in your body mass during COVID-19?	Yes	194	72.9	51	67.1	139	75.1	4	80.0
	Do you feel you have gained body mass?	152	78.4	39	76.5	110	79.1	3	75.0
	*Higher-grade category*	114	80.9	27	81.8	86	81.8	1	50.0
	*Lower-grade category*	21	70.0	7	63.6	13	72.2	1	100.0
Generally, did you get more or less sleep during the stay-at-home periods?	More sleep	111	41.7	29	38.2	80	43.2	2	40.0
	*Higher-grade category*	83	41.7	19	37.3	64	44.1	0	0.0
	*Lower-grade category*	19	52.8	7	43.8	11	57.9	1	100
AUDIT-C Categorised Drinking Risk	Higher risk drinking: Score of 5+ for females, 6+ for males.	107	48.9	39	63.9	65	42.2	3	75.0
	*Higher-grade category*	76	46.9	26	34.2	48	63.2	2	66.7
	*Lower-grade category*	19	63.3	7	36.8	12	63.2	0	0.0
How would you rate your current mental well-being?	Good or very good	112	42.1	38	50.0	73	39.5	1	20.0
	*Higher-grade category*	92	46.2	29	56.9	62	42.8	1	33.3
	*Lower-grade category*	11	30.6	8	50.0	3	15.8	0	0.0
Can you estimate how long you spend on your phone in a typical day?	5+ h	101	38.0	24	31.5	77	41.6	0	0.0
	*Higher-grade category*	79	39.7	16	27.1	60	41.4	0	0.0
	*Lower-grade category*	36	44.4	5	31.3	11	57.9	1	100.0

**Table 5 ijerph-19-16096-t005:** Logistic Regression predicting the likelihood of participants’ self-reported body mass gain during the COVID-19 pandemic (n = 178). Underline indicates a low level of significance (*p* > 0.05).

Independent Variable	B	Sig	OR	95% CI−	95% CI+
Gender *Female* *Male*	−0.07	0.90	1.00 0.94	0.33	2.64
Age group *18–20* *21–23* *≥24*	−0.22 2.09	0.12	1.00 0.81 8.10	0.26 1.83	2.54 35.84
Feelings about body mass *Good/very good* *Neither good nor poor* *Poor/very poor*	−1.00 1.59	0.00	1.00 0.367 4.88	0.11 1.15	1.22 20.68
Perceived BMI Category *Underweight/normal weight* *Overweight/Obese*	1.74	0.03	1.00 5.73	1.25	26.21
Dietary Statement ‘*I sometimes eat because I am bored*’ *Strongly agree* *Agree* *Neither agree nor disagree* *Strongly disagree/disagree*	−1.50 −2.04 −3.75	0.00	1.00 0.22 0.13 0.02	0.07 0.02 0.00	0.74 0.83 0.16
Changes in sleep during stay-at-home period *Stayed the same or don’t know* *More sleep* *Less sleep*	1.86 0.97	0.01	1.00 6.43 2.63	1.97 0.61	20.99 11.39
Sleep quality during stay-at-home period *Good/very good* *Neither good nor poor* *Poor/very poor*	−0.23 −1.49	0.08	1.00 0.80 0.23	0.21 0.06	3.05 0.85
Constant	1.25	0.43	3.49		

## Data Availability

Due to the highly sensitive nature of the material and ethical constraints regarding data sharing and confidentiality, data have not been uploaded to a public repository. Anonymised files can be made available at a later stage upon reasonable request if required.

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
