# Peer review of "The Impact of COVID-19 on the Health-Related Behaviours, Mental Well-Being, and Academic Engagement of a Cohort of Undergraduate Students in an Irish University Setting"

_ijerph, 2022, doi:10.3390/ijerph192316096_

Round 1

Reviewer 1 Report

Dear authors, congratulations on your work. If you will allow me, I would like to make a few contributions.

The objective should be made explicit in the introductory section at the end.

Don't the authors believe that incentivising participation financially for the study may constitute a bias?

Inclusion criteria are not clearly specified in method

The psychometric properties of validated questionnaires should be present in the instruments section.

Implications and limitations of the study are not shown

Author Response

Many Thanks. 

Reviewer 2 Report

Their work regarding the COVID-19 impact on health-related behaviors, mental well-being and academic engagement of college students is very interesting, but needs some improvement before it can be accepted.

0) The theoretical introduction is current to the global COVID-19 environment and the instruments used are presented in detail, which is appreciated.

1) Regarding results, statistical descriptions of the variables and regressions with body mass gain are presented. 

1.1) It is necessary to account for some correlations between the studied phenomena (e.g. methodological: https://doi.org/10.3390/app12157703). Tables 3, 4 and 5 provide many categorical variables that can be correlated.

2) The discussion is very extensive, and tends to exceed the study focus. 

2.1) Summarize this and focus your discussion on relevant findings in comparison to other similar literature.

3) The conclusion can be improved based on the improvement of your results. And focus on correlations rather than descriptions.

Author Response

Many Thanks. 

Round 2

Reviewer 2 Report

The requests for review have been mostly successful.

Author Response

The authors would like to thank the reviewer.